# Comparison of Substance Sources in Experimental Antimicrobial Susceptibility Testing

**Filip Bielec** [1,2,*] , **Małgorzata Brauncajs** [1,2] **and Dorota Pastuszak-Lewandoska** [1]

1   Department of Microbiology and Laboratory Medical Immunology, Medical University of Lodz,
    90-151 Lodz, Poland
2   Medical Microbiology Laboratory, Central Teaching Hospital of Medical University of Lodz,
    92-213 Lodz, Poland
*   Correspondence: filip.bielec@umed.lodz.pl

**Abstract:** Funding is often a constraint when planning research, especially in countries where basic research is underfunded. Researchers must take into account these limitations, e.g., in relation to the selection of appropriate reagents, the source of which may affect the study's final results. The aim of this article was to compare the results of bacteria susceptibility testing using three different sources of antimicrobial: the pure powder available from the supplier and two tablet formulations with different excipients. The chosen substance was furazidin (nitrofuran derivative). The susceptibility was tested on a group of 45 uropathogenic Enterobacterales using both microdilution and disk diffusion methods. The obtained results indicated that despite the relatively higher price, the powder appeared to be the best substance for scientific purposes, especially for quantitative determinations.

**Keywords:** antimicrobial susceptibility testing; microdilution susceptibility test; disk diffusion susceptibility test; furazidin; Enterobacterales

## 1. Introduction

Urinary tract infection (UTI) is defined as the presence of microorganisms in the urinary tract above the internal urethral sphincter, which is usually considered sterile. UTIs are one of the most common bacterial infections globally: it is estimated that about 1/2 of women and more than 1/10 of men experience an episode of UTI in their lifetime [1].

One of the drugs listed in the American [2], European [3], and Polish [4] recommendations for UTI therapy is nitrofurantoin—an antibiotic from the nitrofuran group. Nitrofurans are non-heterocyclic synthetic compounds with antibacterial activity. The bactericidal mechanism of their action is multidirectional consisting of inter alia blocking translation, damaging bacterial DNA, and interfering with cellular metabolic processes. Nitrofurans achieve high urinary concentrations that far exceed therapeutic values, making them effective in treating UTIs. The nitrofurans used to treat UTI include nitrofurantoin, furazidin, and nifurtoinol [5].

Nitrofurantoin is not registered for use in Poland or Russia; instead, furazidin is used as an over-the-counter substitute [6,7]. Unfortunately, furazidin has no standardized in vitro diagnostic susceptibility test, and its susceptibility data is typically extrapolated from nitrofurantoin [4].

Pure furazidin powder is not easily available from popular laboratory reagent suppliers and is relatively expensive, probably due to its uncommon use in research. However, a dozen formulations containing this antimicrobial agent are available over-the-counter in Polish pharmacies at a much lower price for the same mass of furazidin. Despite this, their effectiveness in scientific projects has not been compared in the available literature.

Antimicrobial susceptibility testing includes standardized methods, prepared on the basis of many years of experience and knowledge based on facts. Examples of such standards are recommended by the European Committee on Antimicrobial Susceptibility

Testing (EUCAST)—the microdilution susceptibility test described in ISO 20776-1:2019 [8] and the disk diffusion susceptibility test described in the EUCAST manual [9]. These two methods are the oldest well-established susceptibility diagnostic tests [10]. Different factors can affect the susceptibility testing, these may apply to each step of the procedure used—Table 1 summarizes these factors. Deviation from the standard recommendations while performing susceptibility testing might result in significant changes which can affect the susceptibility interpretation on the resistant–susceptible scale. The consequences may relate to clinical decisions or scientific research results, depending on the type of laboratory performing the method [11–13].

**Table 1.** Factors influencing antimicrobial susceptibility testing.

| Influencing Factors | |
| --- | --- |
| **Directly Related to the Method** | **Indirectly Related to the Method** |
| <ul><li>composition of the medium</li><li>pH value of the medium</li><li>antibiotic concentration and presence of any excipients</li><li>microbial inoculum density</li><li>temperature of incubation</li><li>gas atmosphere of incubation</li><li>length of incubation</li></ul> | <ul><li>correct microbial identification</li><li>selection of appropriate antibiotics</li><li>interpretation following current standards</li><li>laboratory staff experience</li></ul> |

The aim of the present study was to compare the results of Enterobacterales susceptibility testing using three different sources of furazidin: the pure powder available from the supplier (Selleck Chemicals, USA) and two tablet formulations. Our findings may be helpful for providing cost estimates for scientific projects.

## 2. Materials and Methods

Stock solutions of furazidin (5120 μg/mL) were prepared. The tablets were carefully ground in a sterile mortar before dissolution. The excipients used in pharmaceutical tablet formulations are presented in Table 2. The agent was first dissolved in DMSO and then diluted with distilled water. Ultimately, the stock solutions had a DMSO concentration of ~10%. The microdilution susceptibility test with DMSO showed that this concentration did not inhibit the growth of the tested bacteria.

**Table 2.** Excipients used in the tested tablets.

| Tablet 1 | Tablet 2 |
| --- | --- |
| <ul><li>sucrose</li><li>stearic acid</li><li>corn starch</li><li>silica colloidal anhydrous</li></ul> | <ul><li>sucrose</li><li>stearic acid</li><li>potato starch</li><li>lactose monohydrate</li><li>polysorbate 80</li></ul> |

In total, 23 Enterobacterales strains were subjected to furazidin susceptibility testing, including two *Escherichia coli* reference strains (ATCC 25922 and ATCC 8739) and 21 clinical isolates cultured from urine samples obtained between February and March 2021 from the Central Teaching Hospital of the Medical University of Lodz: thirteen *E. coli* strains, seven *Klebsiella* spp. strains and one *Enterobacter* spp. strain (Table S1). All bacteria were stored in Viabank storage beads (Medical Wire & Equipment, Great Britain, Corsham, UK) at −80 °C maximum for six months and regenerated on Columbia Agar with 5% sheep blood (Thermo Fisher Scientific, Waltham, MA, USA), 18–24 h at 37 °C. The susceptibility for furazidin was tested using microdilution and disk diffusion. All determinations were made in triplicate.

The microdilution susceptibility test was performed following ISO 20776-1:2019 [8]. The bacteria were incubated on 96-well titer plates in a series of two-fold dilutions of furazidin (256–0.5 µg/mL) in Mueller–Hinton broth (Thermo Fisher Scientific, Waltham, MA, USA). MIC was defined as the concentration demonstrating a lack of growth according to the EUCAST reading guide for broth microdilution ver. 3.0 [14].

The disk diffusion susceptibility test was performed following the EUCAST methodology, ver. 9 manual [9]. The discs with furazidin were applied to the surface of the inoculated Mueller–Hinton agar plates (Graso Biotech, Starogard Gdański, Poland). The discs used for this test were prepared earlier on that day. Blank discs (Thermo Fisher Scientific, Waltham, MA, USA) were soaked with 19.5 µL of 5120 µg/mL furazidin solution to obtain discs containing 100 µg furazidin. The inhibition zone diameter was measured manually with a caliper according to the EUCAST reading guide for the disk diffusion method [15].

Statistical analysis was performed using Statistica 13 software (TIBCO Software Inc., Palo Alto, CA, USA). The distribution of collected data was checked using the Shapiro–Wilk test. All variables were distributed non-normally. Friedmann ANOVA was used to compare the differences in susceptibility from different furazidin sources. The Wilcoxon test was used for post hoc analyses. The correlations between MICs and inhibition zone diameters were checked using Spearman's test. A *p*-value of 0.05 was considered the limit of statistical significance.

## 3. Results

The obtained raw results of susceptibility testing are presented in the Supplementary Materials (Table S1). As assumed, mean MIC determinations were significantly negatively correlated with the corresponding mean inhibition zone diameters. The different antibiotic sources demonstrated statistically significant differences concerning MICs (Figure 1) and inhibition zones (Figure 2).

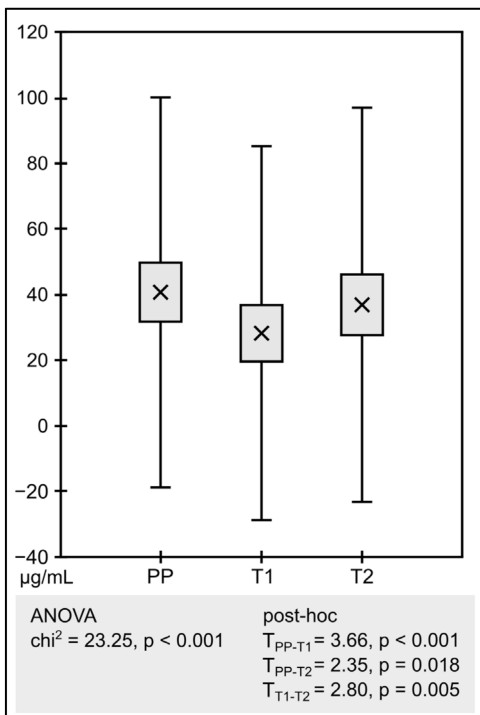

**Figure 1.** Analysis of MICs (PP—pure powder, T1—tablet 1, T2—tablet 2). The graph compares the mean of the determinations. The boxes show standard error and the whiskers shows standard deviation.

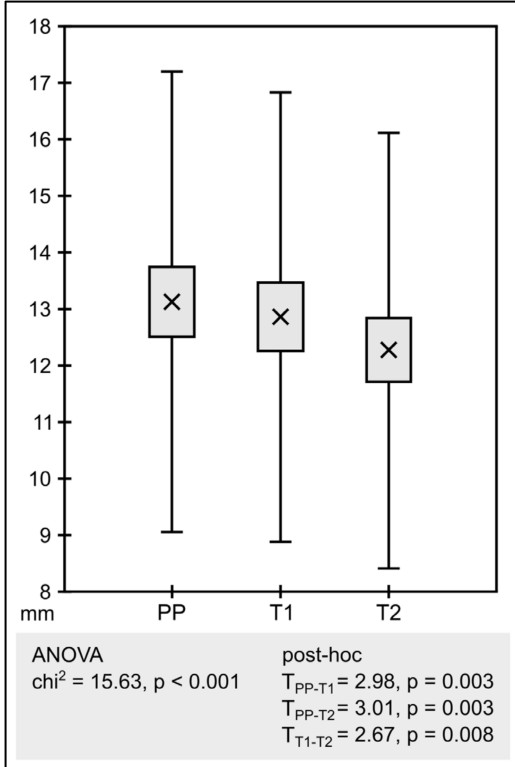

**Figure 2.** Analysis of inhibition zone diameters (PP—pure powder, T1—tablet 1, T2—tablet 2). The graph compares the mean of the determinations. The boxes show standard error and the whiskers shows standard deviation.

The differences obtained were even more evident when the changes in the results were assessed. Tables 3 and 4 show a summary of the changes in MICs and growth inhibition zones depending on the source of the substance tested (from tablets) against the reference method (pure powder), respectively. All data is also presented in the form of a clear graph, visually showing the changes in MICs and growth inhibition zones—Figure 3.

**Table 3.** Summary of changes in MICs depending on the source of the substance tested (T1—tablet 1, T2—tablet 2) against the reference method with pure powder (essential agreement was defined as acquiring an MIC within ±1 doubling dilution away from the reference).

| Substance tested | Number of doubling dilutions away from the reference | | | | | Essential agreement |
| --- | --- | --- | --- | --- | --- | --- |
| | −2 | −1 | 0 | 1 | 2 | |
| T1 | 3 | 16 | 3 | 1 | 0 | 87% |
| T2 | 2 | 7 | 12 | 2 | 0 | 91% |

**Table 4.** Summary of changes in inhibition diameter zones depending on the source of the substance tested (T1—tablet 1, T2—tablet 2) against the reference method with pure powder (essential agreement was defined as acquiring a growth inhibition zone within ±1 mm away from the reference).

| Substance tested | Growth inhibition zone (mm) change from the reference | | | | | | | Essential agreement |
| --- | --- | --- | --- | --- | --- | --- | --- | --- |
| | −3.0 – −2.1 | −2.0 – −1.1 | −1.0 – −0.1 | 0 | 0.1 – 1.0 | 1.1 – 2.0 | 2.1 – 3.0 | |
| T1 | 0 | 1 | 14 | 5 | 3 | 0 | 0 | 96% |
| T2 | 4 | 4 | 9 | 2 | 3 | 0 | 1 | 61% |

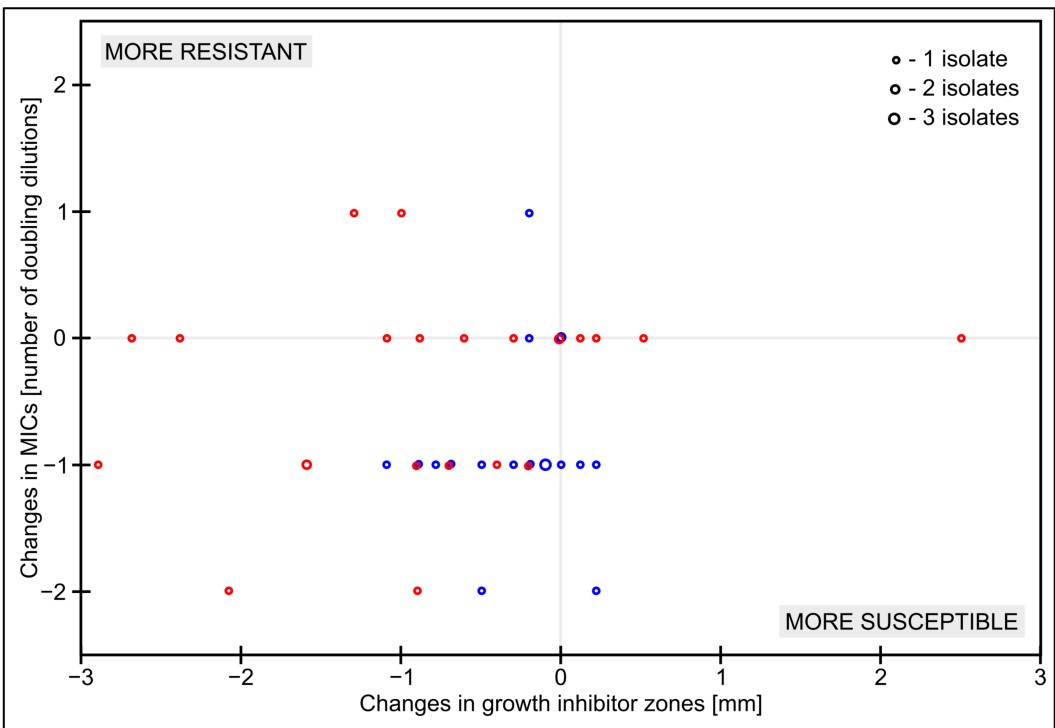

**Figure 3.** Scatterplot of changes in MICs and inhibition diameter zones depending on the source of the substance tested (blue—tablet 1, red—tablet 2) against the reference method with pure powder.

## 4. Discussion

By searching for the factors influencing the results of the antimicrobial susceptibility testing, we could find studies checking the effect of the density of the bacterial inoculum [16], the quality of the used media [17–19], or the quality of the antibiotic discs [20]. We did not find a study similar to ours in the available literature, which was the main reason to take up this topic and share the results with the general public. Perhaps our findings will be useful in science projects conducted outside our department. We also hope that someone will be willing to replicate our study under similar conditions, which would help confirm or question our results.

Smith & Kirby [16] described the so-called inoculum effect, i.e., the influence of the number of bacteria initially inoculated into the assay on the obtained MICs. Higher-inoculum density resulted in higher MICs, and lower-density inoculum resulted in lower MICs.

The standards for antimicrobial susceptibility testing using the disc diffusion method [9] clearly specify the type of medium that is compatible with the method. Unfortunately, for economic reasons, some laboratories do not adhere strictly to this standard. Mohamed et al. [17] showed that the use of media other than those recommended resulted in multiple errors and a high discrepancy in results. On the other hand, laboratories that use the recommended commercial media may be subject to errors due to the poorer performance of some well-known manufacturers' products [18,19]. Concerning commercial antibiotic discs used in the disc diffusion method, Åhman et al. [20] showed various quality issues related to most of the products checked.

In our study examining the influence of different sources of antimicrobial on susceptibility testing, significantly lower MICs were obtained with furazidin derived from tablets. This is most likely due to the action of excipients: their properties may potentially inhibit bacterial growth in an aqueous solution of Mueller–Hinton broth. Both tablets contained sucrose and stearic acid, and their composition differs from other excipients (Table 2). The most significant difference in MICs was observed between tablet 1 and the pure powder. It is possible that the lactose in tablet 2 was an additional nutrient for the bacteria, resulting in higher MICs than for tablet 1.

Likewise, the presence of excipients probably contributed to significant differences in the growth inhibition zones. In this case, however, the opposite effect was observed: the tablets demonstrated smaller inhibition zones than the pure powder. One explanation may be that in this method, the excipients were not added directly to the growth medium but to the paper disc. The water-insoluble excipients (starches, silica, polysorbate) were probably retained on the paper and did not penetrate the Mueller–Hinton agar. On the other hand, water-soluble sugars (sucrose, lactose) could serve as additional nutrients and enhance the growth of bacteria.

Nowadays, we observe a non-stop development of rapid antimicrobial susceptibility testing methods, which is mainly due to the need of diagnosing clinical material from infected patients [21]. New methods are associated with a high risk of errors because they have not yet been sufficiently optimized and standardized—due to their short lifetime [22]. However, it seems reasonable to stick to well-established methods such as microdilution, disk diffusion, or gradient strips in scientific research. The speed of obtaining susceptibility results is not as important in science as obtaining the correct and reliable results. There are reports in the literature on differences in antimicrobial susceptibility testing with newer, automated methods, which may have a significant impact on the correct interpretation of the results [23,24].

### 5. Conclusions

The validity of antimicrobial susceptibility testing depends on every feature of the test, including the factors listed in Table 1. It is important to understand the impact of different variables on the test. Any excipients present with the substance tested may have an influence on the final susceptibility result.

To quantitatively determine the susceptibility of the microorganism to an antimicrobial substance, it is necessary to use standardized pure reagents with the appropriate characteristics. Financial constraints should not play a role.

However, in cases where only qualitative observation is sufficient, e.g., a synergy study, it is possible to use substances from other sources in which excipients may be present (e.g., from available pharmacotherapeutic agents). Such a procedure could effectively minimize the costs of research.

**Supplementary Materials:** The following supporting information can be downloaded at: https://www.mdpi.com/article/10.3390/scipharm91010010/s1, Table S1: The data presented in the study—results of furazidin susceptibility testing.

**Author Contributions:** Conceptualization, F.B. and D.P.-L.; methodology, F.B.; validation, M.B.; investigation, F.B.; resources, M.B. and D.P.-L.; data curation, F.B.; writing—original draft preparation, F.B.; writing—review and editing, F.B., M.B. and D.P.-L.; supervision, M.B. and D.P.-L.; funding acquisition, D.P.-L. All authors have read and agreed to the published version of the manuscript.

**Funding:** This research was funded by Medical University of Lodz.

**Institutional Review Board Statement:** Not applicable.

**Informed Consent Statement:** Not applicable.

**Data Availability Statement:** The data presented in this study are available in Supplementary Materials (Table S1).

**Conflicts of Interest:** The authors declare no conflict of interest.

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
