# Peer review of "Comparison of Substance Sources in Experimental Antimicrobial Susceptibility Testing"

_scipharm, doi:10.3390/scipharm91010010_

Round 1
Reviewer 1 Report
The authors present a study on the differences in quantitative antibiotic susceptibility testing using antibiotics from different sources.
Generally I like these types of articles, the authors show differences in using pure antibiotic powder compared to cheaper versions in pill form with added excipients. AST is a standardised test and small differences in any factor (including just lab to lab with the same reagents) have an impact. How much of an impact is important to understand. Some laboratories have used other media formulations, due to cost, and the differences can be dramatic.
I would like the authors to touch on some literature showing differences in standard testing conditions can affect the outcome, especially if this includes some of the excipients used in this study. If this comes down to word limit, I would prefer an emphasis on this in place of the use case in UTIs. I see this as an AST study comparing methods/reagents that is potentially relevant to other antibiotics than the current introduction focusing on the use in UTIs.
I think the data should be presented in another way. Usually, the index test (the antibiotic from the pill form) is tested against a reference method (pure antibiotic) and the fold change from the reference method is plotted. This normalises the data and will make the changes in MIC more obvious. i.e the reference method (pure antibiotic) is not plotted because it will be 0.
The same can be done for Table 1 – instead of listing each isolate, normalise and plot the change in MIC from the reference test. Below is an example but is not mandatory. Table 1 can stay as is if the authors prefer but the data in the plots should be presented as fold change.
Broth microdilution |
Number of doubling dilutions away from the reference |
Essential agreement |
||||
|
-2 |
-1 |
0 |
1 |
2 |
|
F2 |
3 |
15 |
3 |
|
|
78% |
F3 |
|
|
|
|
|
|
|
|
|
|
|
|
|
I agree with your conclusion concerning the exipients in liquid and solid media tests.
Author Response
Thank you kindly for your interest in the article and sending valuable comments and tips - including very detailed ones. It was extremely helpful. We took into account all comments.
In the introduction, a paragraph and a table on factors influeacing the AST have been added.
The results section has also been modified. We decided that the table with the list of all indications will eventually find a place in the supplement. However, we decided to leave graphs comparing the averages of the obtained determinations. In addition to them, we have added tables and figures suggested by the reviewer - now all the results and their tendencies are clearly visible.
Reviewer 2 Report
scipharm-2121026
In this manuscript presented in form of 'Communication', a study was conducted in order to to compare the results of Enterobacterales susceptibility testing using three different sources of furazidin, acting as antimicrobial agent, in three forms: the pure powder available from the supplier and two tablet formulations with different excipients. The aim of the investigation was find evidence about cost estimates for scientific projects.
Results showed that even though at a relatively higher price, the pure powder resulted the best substance for scientific purposes, especially for quantitative determinations.
The manuscript is interesting and offers a point of view that can be useful for conducting analyses in laboratory giving optimum in terms of results and of costs.
The Abstract is not very clear in some points, at line 16 rephrase the sentence.
Table 1 is reported in the Introduction section, but it is well adapt for the materials and Method section.
Lines 59-62: change the species names (Escherichia coli, Klebsiella and Enterobacter) to Italic style. Instead, leave 'spp.' Not in Italic style.
Results should need some more description.
Author Response
Thank you for your interest in the article and sending valuable comments and suggestions. It was extremely helpful. We took into account all comments.
The indicated sentence in the Abstract has been rephrased. Other indicated spelling and stylistic errors have also been corrected.
We moved the table with the description of the excepients in the tablets to the Methods section - it definitely fits better there.
In addition, the Results section has been completely revamped following the advice of another reviewer.